# Microbial Enhancement of Plant Tolerance to Waterlogging: Mechanisms and Interplay with Biological Control of Pathogens

**DOI:** 10.3390/ijms26168034

**Published:** 2025-08-20

**Authors:** Tomasz Maciag, Dorota M. Krzyżanowska

**Affiliations:** 1Laboratory of Phytopathology, Department of Plant Protection, The National Institute of Horticultural Research, Konstytucji 3 Maja Street 1/3, 96-100 Skierniewice, Poland; 2Laboratory of Plant Microbiology, Intercollegiate Faculty of Biotechnology of the University of Gdansk and the Medical University of Gdansk, University of Gdansk, Antoniego Abrahama Street 58, 80-307 Gdansk, Poland; dorota.krzyzanowska@biotech.ug.edu.pl

**Keywords:** PGPM, PGPB, abiotic stress, plant growth promotion, hypoxia

## Abstract

Climate change causes major agricultural losses, driven both by the rise of plant diseases and by extreme weather events such as droughts and floods. Increased precipitation can lead to waterlogging of important crops. The roots of plants submerged in water have limited access to oxygen, which leads to hypoxia, which, in turn, reduces plant resistance to other factors, e.g., plant pathogens. On the other hand, beneficial microorganisms can help plants oppose abiotic stress, e.g., by producing plant hormones or osmoprotectants such as trehalose, to increase plant tolerance to drought. It turns out that plant-beneficial microorganisms can also increase plant resistance to waterlogging. This can be achieved by various mechanisms that involve the production of 1-aminocyclopropane-1-carboxylate (ACC) deaminase, which reduces the amount of ethylene accumulated in the submerged roots. This can stimulate the production of reactive oxygen species scavengers that protect plants from the oxidative stress caused by less efficient anaerobic metabolism, produce plant hormones that help plants to better adapt to low-oxygen conditions, and shape the plant microbiome, supporting plant growth in waterlogging conditions. This review outlines plant responses to waterlogging and discusses examples of microorganisms that improve plant tolerance, focusing on their underlying mechanisms.

## 1. Introduction

The global food demand is projected to increase by 35–56% between 2010 and 2050, a range that remains relevant as of 2025 due to the ongoing population growth [1]. A proportional growth in production needs to be achieved despite the limited availability of agricultural land, which is further strained by the pressure from growing cities, soil pollution, and the cultivation of energy crops [2]. This global problem of an impending hunger catastrophe has been underscored by more than 150 Nobel and World Food Prize laureates, who signed an open letter in January 2025 calling for immediate action [3]. On top of that, there is a growing threat from spreading plant pathogens and devastating weather conditions caused by climate change [4]. This situation calls for a swift reply not only to increase farmland productivity and mitigate the losses but also to minimize the negative impact of agricultural practices on the environment [4]. To achieve this ambitious goal, a multidisciplinary approach is essential, integrating efforts in plant protection, breeding, logistics, and legislation [2].

One of the most pressing problems caused by climate change in modern agriculture is flooding, with its devastating effects on crop production [5]. Plant flooding induces hypoxia in roots, leading to partial death of roots and consequently deteriorated growth, which causes decreased crop yield and quality [6]. A promising approach to reducing the harmful effects of this factor is to use microorganisms that naturally occur on plants. A plant and its associated microorganisms together form a unified biological system known as a holobiont, which functions as an integrated unit that is capable of resisting the adverse effects of the surrounding environment [7]. This enduring, dynamic collaboration forms the foundation of plant health, enhancing resilience against biotic stresses, such as pathogens and pests, and abiotic challenges, including drought, flooding, and soil toxicity [8].

Microorganisms constituting the holobiome can exert diverse effects on the plant. Plant-Growth-Promoting Microorganisms are a group of microorganisms, including rhizospheric (e.g., *Bacillus*) and endophytic (e.g., *Pseudomonas*) bacteria and fungi (e.g., *Trichodrema*), that are beneficial for plant health [9]. PGPMs can support plant growth through various modes of action, including phosphate solubilization, nitrogen fixation, increasing the soil water-holding capacity, hormonal growth stimulation, protection from pathogens through direct antagonism or competition for environmental niche, or induction of plant resistance to pathogen invasion (via Induced Systemic Resistance, ISR) [10]. Just as they increase resistance to pathogen-mediated biotic stress through ISR, PGPMs can improve plant tolerance to abiotic stress through Induced Systemic Tolerance (IST) [11]. IST is a relatively new term that was introduced to underline the positive impact of microorganisms on plants’ ability to defy unfavorable environmental conditions, particularly drought and salinity stress [11]. IST, similarly to induced systemic resistance ISR, relies on the production of plant hormones, stimulating similar physiological and metabolic changes in plants to support plant survival [11].

Some mechanisms fundamental to the enhancement of plant tolerance to abiotic stress closely resemble those involved in inducing plant resistance to biotic factors. For example, the thickening of outer apoplastic barriers, inter alia, via increased suberization provides improved plant tolerance to both biotic and abiotic stress [12]. In addition, plants use similar hormonal signaling pathways to turn on their defenses against biotic and abiotic stress factors [13], and they produce secondary metabolites with dual functions, both against pathogens, thus protecting plant macromolecules against physical factors such as UV [14]. Nonetheless, there are key differences worth emphasizing [15]. Firstly, plants can never achieve full resistance to abiotic stress, since it is a physical or chemical factor, and the tolerance is always qualitative. For example, plants can be more tolerant of drought, but they cannot grow without water. In opposition to that, certain plants can be or become resistant to given pathogens through the presence of a particular version of a gene that a specialized pathogen cannot target. This phenomenon is most evident in the gene–for-gene scenario, where plant resistance to a given pathogen is determined by the presence or absence of a distinct resistance gene targeted against the pathogen’s avirulence gene [16]. The latter model is characteristic of specialized biotrophic pathogens that coexist with the plant, hindering its growth and suppressing the activation of its resistance mechanisms [17]. The resistance to necrotrophic pathogens, which take the resources from decaying plants, tends to be more quantitative than qualitative, where absolute resistance is rarely achieved and is dependent on the expression of sets of genes and environmental factors [17]. For instance, in the case of broad-host-range necrotrophic bacteria, such as the Soft Rot *Pectobacteriaceae,* the disease resistance would be more qualitative and rely on physiological adaptations rather than the expression of single genes conferring resistance, which is more similar to the tolerance to abiotic stress [18,19,20,21].

Biological plant protection (biological control) refers to using living organisms to protect plants from pests, diseases, and weeds [22]. However, it is essential to consider a fundamental concept in plant pathology known as the Disease Triangle. According to this principle, disease symptoms in plants develop when a virulent strain encounters a susceptible host under disease-favorable conditions, though the impact of each factor varies depending on the disease [23]. Plant pathogens generally take advantage of plant weakening by abiotic stress [18,19,20,21,24]. For example, Soft Rot *Pectobacteriaceae* can sense the decrease in plant defenses and favorable environmental conditions caused by high moisture and hypoxia, responding with induced production of virulence factors and disease onset [21]. *Pectobacterium* spp. can detect fragments of degraded pectins using chemoreceptors to induce virulence against a weakened host [25]. Moreover, these bacteria can sense hypoxia using the global regulator c-di-GMP, which upregulates the production of virulence factors [26]. In another example, *Pseudomonas syringae* produces ice nucleation proteins, decreasing plants’ tolerance to low temperatures and leading to microfractures in plant tissue, which provide entry points for bacteria [24]. Therefore, although abiotic and biotic stresses pose different challenges to plants, they often coexist and require a synchronized response [27]. This situation adds complexity to the problem of plant protection against diseases and abiotic stress [28]. However, it also provides hope for developing strategies to fight the negative influence of combined stress factors [4,15]. For instance, the ISR response involved in disease resistance can use the same signaling molecules as IST, such as ethylene, and it can be induced by the same factor, such as PGPMs [29]. Due to an overlap between plant responses to biotic and abiotic stressors, the same mechanisms can be used to protect plants from physicochemical factors and plant pathogens. Plant protection against biotic and abiotic stresses is, however, subjected to different legal regulations, as the former traditionally involves harmful chemicals targeting plant pests and pathogens, while the latter relies on fertilizers, which help plants to tolerate physical problems such as increased or decreased water availability. Nevertheless, if we use mechanisms that increase plants’ tolerance to abiotic stress, we can simultaneously increase their resistance to pathogens associated with this environmental factor.

In this review, we focus on the mechanisms used by beneficial microorganisms to enhance plant tolerance to waterlogging while highlighting their interplay with biological control. Waterlogging is a process of temporary saturation of soil with water, which disturbs the gas exchange in roots, leading to their necrosis. When discussing waterlogging-protection mechanisms, we structure the review around distinct aspects of microbial influence, highlighting how they intertwine and providing context for understanding their interconnection. Although much scientific interest has been focused on the use of microorganisms to mitigate the negative impact of biotic or abiotic stress, such as drought, the mitigation of waterlogging is much less studied, and only a limited number of microbial strains have been proven to increase plant tolerance to waterlogging [30]. This review aims to present the current findings on the use of microorganisms to alleviate waterlogging stress in agriculture and investigate the mechanisms of their activity to aid the future development of microbial formulations that are effective against this growing threat [5].

## 2. Plant Response to Waterlogging

Waterlogging stress in plants arises when excess water accumulates in the soil, typically due to flooding, saturating the root zone and limiting oxygen availability for plant roots. It occurs when the soil is unable to drain excess water efficiently, leading to hypoxic (low oxygen) or anoxic (no oxygen) conditions that negatively impact plant growth and soil health [31]. By definition, waterlogging refers to a situation in which only the soil and underground parts of plants are submerged in water—when the water level advances above the ground level, the process is called submergence [32].

The fact that during waterlogging, water blocks the gas exchange in roots not only poses a threat to field-grown crops but is also a limiting factor for hydroponic agriculture [33]. Hydroponically grown plants, due to their stable environment, can develop roots adapted to flooding, and they do not suffer from oxidative stress caused by sudden decreases or increases in oxygen concentration resulting from the changing water levels. Despite that, many plants require the optimization of hydroponic culture conditions for decreased oxygen exchange through water [34].

Plants are not defenseless against changing weather conditions and the associated abiotic stress. In response to waterlogging, they exhibit a range of adaptive strategies involving both metabolic and anatomical mechanisms. Metabolically, plants switch to anaerobic or fermentative pathways [35,36], suppress their secondary metabolism [37], detoxify reactive oxygen species (ROSs) via enzymatic and antioxidant systems [38], and reduce their overall metabolic rate to minimize oxidative damage [36,39]. Anatomical adaptations include the formation of adventitious roots near the soil surface [40], development of aerenchyma to facilitate internal oxygen transport [41,42], thickening of root cell walls to limit oxygen loss [43], and, in some wetland species, enhanced wax deposition on leaf surfaces [44] and internode elongation to reach the water surface [45] (Figure 1). A comprehensive review of plant responses to waterlogging is available in a study by Daniel and Hartman (2024) [46]. However, as the mechanisms by which microorganisms alleviate waterlogging stress are tightly connected to specific plant response pathways, each microbial mechanism discussed is accompanied with a concise summary of the relevant plant responses.

## 3. Microbial Mechanisms Enhancing Plant Tolerance to Waterlogging

Plants do not stand alone in their efforts to protect themselves against the influence of abiotic stress factors. The beneficial bacteria within the plant microbiome perform and trigger different metabolic processes that contribute to a plant’s resistance [7]. The following review summarizes known mechanisms by which microorganisms alleviate stress specifically induced by waterlogging, organized according to the metabolic processes they affect. As this is a relatively new area of research, with limited data available specifically on microbial mechanisms under waterlogging stress, the review includes both mechanisms identified in PGPR tested directly under waterlogging conditions and, for a broader perspective, microorganisms studied more generally for stress alleviation, whose traits overlap with those involved in mitigating waterlogging stress.

### 3.1. Targeting Ethylene Synthesis

Ethylene is the major plant hormone responsible for waterlogging detection and signaling. It is synthesized in plants from the amino acid methionine, which is first converted into its derivative, S-adenosyl-L-methionine (SAM). SAM is then transformed into 1-aminocyclopropane-1-carboxylic acid (ACC) by ACC synthase, and finally, ACC is oxidized by ACC oxidase to produce ethylene, along with byproducts such as CO_2_ and cyanide [47] (Figure 2). Ethylene is constantly produced in plant roots and diffuses through the plant cell wall into aerated soil. Nonetheless, during flooding, water fills the air pockets in the soil, blocking the gas exchange between the plant and the atmosphere, leading to an increased concentration of ethylene in the flooded roots [48].

Ethylene plays a pivotal role in plant adaptation to waterlogging by inducing anaerobic metabolism, which allows the plant to preserve oxygen [58] (Figure 1). Moreover, plants use ethylene signaling to develop physiological adaptation to waterlogging, including lysogenous aerenchyma formation. In this process, in response to the increased concentration of ethylene, cortex cells undergo programmed cell death and produce cell wall lytic enzymes, leading to the formation of an air-filled space, which can later be used for gas transport to the submerged parts of the roots [59]. Ethylene also stimulates the formation of adventitious roots through both auxin-dependent and auxin-independent mechanisms [29]. The new roots are formed closer to the water surface to compensate for the loss of roots caused by hypoxia [60] (Figure 3).

Although ethylene signaling in plants aims to increase plant resistance to hypoxia caused by waterlogging, the extent of ethylene production may lead to chlorosis and, consequently, necrosis of plant tissues [61]. Therefore, when a plant is subjected to a sudden increase in water level without being previously primed, the accumulated ethylene concentration will have a deleterious effect on the plant’s growth [62] (Figure 1). Consequently, although ethylene plays a central role in plant adaptation to waterlogging, its accumulation must be tightly regulated by other plant hormones to prevent adverse effects on growth [45] (Figure 4).

Numerous PGPMs are reported to increase plant resistance to abiotic stress due to the production of ACC deaminase, an enzyme known to lower ethylene levels by breaking down the immediate precursor of ethylene in plants [63]. Multiple microorganisms can reduce the amount of ethylene accumulated in the submerged plant roots due to the production of 1-aminocyclopropane-1-carboxylate (ACC) deaminase, which stimulates plant tolerance to waterlogging [30]. Bacterial ACC deaminase breaks down ACC—the immediate precursor of ethylene in plants [63]. As a result, it decreases the amount of ACC available to the ACC oxidase, a plant enzyme responsible for ethylene synthesis, leading to a decreased concentration of ethylene [54] (Figure 2). Due to the promising potential of this mechanism of increasing plant tolerance to abiotic stress, it is currently the best-studied mechanism of alleviation of waterlogging stress by microorganisms [30] (Table 1). Furthermore, a substantial body of research exists on ACC-deaminase-producing microorganisms in the context of induced plant tolerance to drought and salinity stress [63]. This suggests that similar mechanisms may overlap and that strains producing ACC deaminase could also hold potential for mitigating waterlogging stress. For example, *Streptomyces* sp. GMKU 336 was selected for its ability to suppress salinity stress in rice by ACC deaminase [64], and it can also mitigate waterlogging stress in mung bean via the same mechanism [65] (Table 1). However, the effectiveness of these strains still requires experimental verification under waterlogging-specific conditions.

Some microorganisms are also known to have the opposite effect—namely, increasing ethylene levels in plants. Ethylene can be synthesized from methionine [77]. However, microorganisms that use methionine as a substrate for ethylene production have a significantly lower impact on overall ethylene levels compared to those that utilize alternative substrates. This is because, although they contribute to ethylene synthesis, they also draw on methionine—a key precursor in plant ethylene biosynthesis—thereby competing with the plant for the same limited resource. As opposed to that, the plant pathogen necrotrophic *Penicillium digitatum* produces ethylene from glutamic acid and in quantities that far exceed what the fruit itself naturally produces during ripening [78]. Increasing the concentration of ethylene in plant tissues can facilitate pathogen invasion by weakening plant defenses, such as through the suppression of the secondary metabolism or inhibition of ROS bursts, which can otherwise trigger a hypersensitive response and necrosis, aimed at limiting the pathogen’s spread [29]. Conversely, ethylene can reduce the virulence of certain plant pathogens; for example, under elevated ethylene conditions, necrotrophic *Botrytis cinerea* exhibits decreased pathogenicity in *Nicotiana benthamiana* Domin [79].

### 3.2. Targeting Hormonal Signaling

Although ethylene plays a primary role in sensing waterlogging conditions, an appropriate plant response is achieved through the involvement of other hormones that help mitigate the negative effects of excessive ethylene accumulation [45] (Figure 2). One of the hormones [29,61] acting downstream of ethylene is auxin. Ethylene regulates the production and transport of auxins to the flooded parts of the plant, promoting cell division and the formation of new roots to help plants with respiration under submerged conditions [60]. Plant-beneficial microorganisms can metabolize the L-tryptophan exudated by plant roots and convert it into auxin indole acetic acid (IAA), promoting advantageous root formation (Figure 5). Surprisingly, IAA is commonly produced by different rhizobacteria, including phytopathogens, which tend to use different enzymatic pathways [80]. The phytopathogens tend to use the indole-3-acetamide pathway, while plant-beneficial bacteria most commonly use the indole-3-pyruvic acid pathway [81]. Some microorganisms, such as *Azospirillum brasilense* [82] or *Micrococcus aloeverae* [83], can also use the tryptophan-independent pathway, which has not been fully described yet [81]. For example, IAA-producing *Klebsiella variicola* AY13 increases soybean tolerance to waterlogging stress and increases shoot length, root length, and chlorophyll content of the submerged plants [84]. A study conducted on a pineapple aimed at the isolation of plant-beneficial bacteria for increasing plant tolerance to drought showed a high level of cooccurrence of auxin production by the isolates with the production of ACC deaminase [85]. This is in line with the prevalence of IAA-producing strains among ACC-deaminase-producing isolates that enhance waterlogging tolerance in plants, as summarized in Table 1.

Another plant hormone involved in plant water stress response is abscisic acid [86]. This important plant hormone is responsible for the control of stomata closure, regulating plant water potential, and suppressing the adventitious root formation induced by IAA [87]. It has been shown that plant-beneficial rhizobacterium *Rhodobacter sphaeroides* KE149 increases soybean tolerance to waterlogging by reducing the concentration of abscisic acid while increasing the concentration of jasmonic acid [88]. Jasmonic acid is a plant hormone that is most widely studied for its role in Induced Stress Resistance (ISR). Beneficial microorganisms present microbe-associated molecular patterns to plants, which leads to the induction of plant stress response and an increased concentration of jasmonic acid, which, in turn, stimulates the plant defenses against necrotrophic pathogens [89]. It has been shown that exogenous application of methyl jasmonate—a methylated derivative of jasmonic acid—induces waterlogging tolerance in pepper [90]. However, data concerning the role of jasmonates in plant fitness and disease resistance are not easy to interpret due to their dual role in plant metabolism. On the one hand, jasmonic acid induces rice resistance to insect attack by water weevil, *Lissorhoptrus oryzophilus,* and on the other hand, it reduces rice growth and yield [91]. Moreover, the plant-protective effect is not universal—while jasmonates generally increase pine resistance to insect attacks by the pine weevil *Hylobius abietis*, they also increase pine susceptibility to the blue-stain fungus *Endoconidiophora ponecessarylonica* [92]. Additionally, plant-pathogenic *Fusarium oxysporum* produces jasmonates to suppress plant resistance mechanisms to facilitate its invasion [93]. However, in a different study, it has been shown that methyl jasmonate produced by *Trichoderma harzianum* increases resistance to *Fusarium chlamydosporum* wilt in tomato [94]. Generally, jasmonates promote plant resistance against necrotrophic and hemibiotrophic pathogens and increase tolerance to numerous abiotic stresses, although they induce plant senescence and growth arrest [95]. The fact that some strains of the necrotrophic pathogen *Fusarium chlamydosporum* have overcome jasmonate-induced defenses and produce jasmonates to suppress salicylic-acid-based defenses underlines the necessity of the coexistence of these two alternative plant defense mechanisms [94].

The production of jasmonic acid is under the control of another plant hormone that is responsible for plant resistance to stress—salicylic acid. Those hormones act antagonistically towards each other, and salicylic acid is considered to play a major role in plant response to biotrophic pathogens [96]. Salicylic acid can be produced by both plant-beneficial [97] and plant-pathogenic [98] microorganisms. This plant hormone is important for plant adaptation to waterlogging, as it stimulates the formation of adventitious roots and aerenchyma [99].

Gibberellins are responsible for plant internode elongation, helping rice escape waterlogging [98]. It has been shown that the exogenous application of gibberellins increases mung bean growth and chlorophyll content, and it decreases the concentration of reactive oxygen species during waterlogging [100]. Although gibberelins seem to play a beneficial role in plant response to waterlogging, they are also produced by plant-pathogenic fungi *Gibberella fujikuroi* [101] and bacteria [102], playing a role as their virulence factor. Contrariwise, the gibberellin-producing bacterium *Bradyrhizobium diazoefficiens* positively regulates soybean nodule size [103]. Moreover, the plant-growth-promoting bacteria *Priestia aryabhattai* and *Pseudomonas frederiksbergensis* increase broccoli and mallow germination due to gibberellin production [104]. Although both plant pathogens and plant-beneficial microorganisms often possess a gibberellin biosynthesis operon, symbiotic bacteria often lack the gene responsible for one of the final steps of gibberellin synthesis [105]. This indicates that the overproduction of gibberellins may have harmful effects on plants, and the exogenous production to promote plant growth should be under partial control of the host plant, meaning that at least one step of gibberellin production has to be performed by the host enzyme.

Another important plant hormone with a significant yet relatively poorly understood role in plant waterlogging tolerance is melatonin [106]. Melatonin was shown to amplify waterlogging resistance in apple by scavenging reactive oxygen species and downregulating ethylene levels [107]. Melatonin also suppresses the anaerobic metabolism, reducing ROS production, and, together with dopamine, it plays an important role in recruiting plant-beneficial microorganisms that help the plant to defy waterlogging stress [108] (Figure 2). Dopamine, similarly to melatonin, contributes to reactive oxygen species scavenging, minimizing the negative impact on plant metabolism [109].

### 3.3. Targeting Metabolism and Reactive Oxygen Species

Plant hormonal signaling allows plants to adapt to changing environmental conditions metabolically and anatomically [110,111]. Waterlogging severely disrupts root respiration due to restricted gas exchange [112]. To manage oxygen scarcity, plants switch to less energy-efficient anaerobic metabolism [35]. Despite the limited availability of oxygen, anaerobic metabolism under hypoxic conditions leads to the accumulation of reactive oxygen species (ROSs)—a seemingly counterintuitive outcome explained by several mechanisms that promote ROS generation even in low-oxygen environments. Firstly, anaerobic metabolism is less efficient than aerobic metabolism and, therefore, requires more cycles to produce the same amount of energy as aerobic metabolism would [113]. Secondly, during anaerobic metabolism, energy production relies on the electron transport along the mitochondrial membrane, which causes sporadic electron escape [113]. Additionally, although the concentration of oxygen is lower in hypoxia, this non-polar gas tends to accumulate inside hydrophobic cell membranes, including the mitochondria, which promotes the creation of reactive oxygen species by escaping electrons [114]. Furthermore, during hypoxia, plants utilize lipids from cell membranes for energy and suppress their secondary metabolism to limit energy consumption, which weakens cell membranes, promoting electron leakage [37]. To minimize ROS toxicity, plants have to reduce their metabolic rate during flooding periods [36]. Amidst hypoxia, wheat and maize rely on fermentative metabolism, while rice switches to metabolic dormancy, therefore limiting the negative impact of hypoxia [36].

ROSs can be partially detoxified by enzymatic processes and antioxidants [38]. While the production of ACC deaminase and reducing ethylene levels are the best-known mechanisms of Induced Systemic Tolerance (IST) of plants for waterlogging, PGPMs usually promote plant growth by incorporating multiple modes of action [115] (Table 1). This additional effect is frequently associated with the microbial production of compounds that help lower ROS levels in plants. For example, *Achromobacter xylosoxidans* can enhance proline accumulation in holy basil (*Ocimum tenuiflorum* L.) [50]. Proline is an important plant hormone overproduced by plants under abiotic stress [116,117]. Proline has a positive impact on plant recovery from salinity, drought, waterlogging, and metal pollution stress due to its metal-chelating properties, scavenging of reactive oxygen species, and action as a signal molecule [118]. It has been shown that the arbuscular mycorrhizal fungus *Funneliformis mosseae* can reduce the negative impact of waterlogging on peach (*Prunes persica* (L.) Batsch) seedlings due to the increased accumulation of proline and chlorophyll in plant tissues [119].

Melatonin, another plant hormone and metabolite, also helps plants withstand waterlogging stress. The molecule not only deactivates reactive oxygen and nitrogen species but also serves as a crucial signaling molecule, linking the synthesis and transport of indole-3-acetic acid (IAA) with signaling molecules involved in plant development and abiotic stress sensing, such as ethylene and nitric oxide [120]. It has been shown that endophytic bacteria isolated from the roots of grapevine and belonging to the *Agrobacterium*, *Pseudomonas*, and *Bacillus* genera could produce melatonin from L-tryptophan, thus supporting plant tolerance to abiotic stress [121]. The role of melatonin does not end there. Melatonin, together with dopamine, decreases waterlogging stress in apples by scavenging reactive oxygen species and recruiting plant-beneficial endophytes such as *Cellvibrio*, *Novosphingobium*, and *Propionivibrio* [108]. Dopamine, another important metabolite derived from the plant metabolism of L-tryptophan, plays a key role in alleviating abiotic stress in plants. The synthesis of this compound in *Cannabis sativa* L. was shown to be enhanced by the following endophytic bacteria: *Serratia marcescens* and *Enterobacter cloacae.* These endophytes convert L-tryptophane into L-dopa—a direct precursor of dopamine [122]. The exogenous application of dopamine to apple trees promotes the growth of the following beneficial endophytes: Candidatus_*Kaiserbacteria*, *Humicola*, *Hydrogenophaga*, *Methyloversatilis*, and *Simplicispira*, which further help the host plant defy waterlogging stress [108].

### 3.4. Targeting the Plant Anatomical Changes

More efficient metabolisms during extended periods of waterlogging can be achieved thanks to anatomic adaptations, which can help plants more efficiently acquire and transport oxygen to the submerged parts of a plant [123]. This can be accomplished by decreasing the oxygen loss due to reduced root surface area, increased cell wall thickness, elevated oxygen uptake through new roots developed closer to the soil surface, and facilitated oxygen transport through aerenchyma-air-filled spaces inside the roots [86].

Flooding-caused hypoxia leads to the death of plant roots, which have to be regenerated to sustain plant metabolism. Hormonal signaling allows a plant to develop new adventitious roots at the stem base closer to the soil surface, where oxygen is more abundant [40] (Figure 3). The newly formed roots are shorter and thicker to minimize radial oxygen loss through the root surface into the soil [41]. The root cell wall, heavily packed with suberin and callose that form a radial oxygen cell wall barrier, further reduces the gas leakage from the limited root surface area into the surrounding environment [43]. It is worth pointing out that the thickened cell wall barrier limits the nutrient and oxygen uptake through the roots, especially under normoxic conditions following waterlogging [124].

Underneath this barrier, ethylene and abscisic acid induce the formation of lysogenous aerenchyma. This type of plant tissue, characterized by air-filled spaces, enhances gas exchange and aids plant tolerance to waterlogged or low-oxygen conditions by facilitating oxygen transport from the shoots to the roots. It is commonly found in wetland plants, where it plays a crucial role in mitigating the effects of flooding by maintaining root respiration even under poor soil aeration [42]. In wetland species, aerenchyma formation does not require induction by ethylene, but its volume increases after waterlogging [41]. Additional adaptations to waterlogging in wetland species can also be found in the aboveground parts of the plant. In submerged rice, excessive wax deposition, induced by the wax synthesis gene LGF1, on the leaf surface helps to retain an air layer and helps the leaves to escape hypoxia [44]. The LGF1 gene expression is controlled by vascular plant one zinc-finger (VOZ) transcription factors, which are predicted to be regulated by auxins, abscisic acid, and methyl jasmonates [125]. Moreover, when the aerial part of the plant is underwater, the gibberellins promote internode elongation, allowing plants to reach the water surface to acquire oxygen [45].

Beneficial microorganisms have been reported to stimulate plants in ways that alleviate the resulting adverse effects of waterlogging by stimulating them to undergo favorable anatomical changes. For example, *Pseudomonas putida* KT24440 causes the redistribution of hairy roots of *Vallisneria natans* to minimize the radial oxygen loss, maximizing nutrient adsorption [126] (Figure 5).

It has been shown that *Bacillus subtilis* and *Pseudomonas mandelii* promote wheat cell wall suberization and lignification, inducing the formation of apoplastic barriers [127]. Plant cell wall suberization leads to the formation of a radial oxygen loss barrier, one of the most important plant morphological adaptations to waterlogging [43].

It has been shown that the water plants *Halophila ovalis* and *Zostera muelleri* can use a radial oxygen loss barrier to modulate the spatial distribution of root-inhabiting bacteria to facilitate sulfide oxidization of sediments [128]. The majority of the root systems in these plants are protected from oxygen loss and sulfide using a radial oxygen loss barrier. Only the root tip cell walls contain less suberin and allow for gas exchange; as a result, the root tips are colonized by sulfide-oxidizing bacteria, which detoxify the surrounding root environment. Meanwhile, the majority of roots are colonized by sulfide-reducing bacteria [128]. Although this phenomenon has not been extensively studied in terrestrial plants, this example illustrates how morphological adaptations typical of waterlogged conditions can potentially influence the spatial distribution of the members of the microbiome.

Furthermore, *Bacillus altitudinis* increases root thickness in rice plants [129]. The root thickness in rice is another important aspect of adaptation to waterlogging, as it helps plants reduce radial oxygen loss and transport oxygen to the submerged part of the roots [130]. The formation of aerenchyma—air-filled spaces within plant roots that facilitate oxygen transport—results from apoptosis of specific root cells, a process mediated by ethylene [110,111]. The formation of aerenchyma in corn, stimulated by the plant-growth-promoting bacterium *Paenibacillus lentimorbus,* increases plant tolerance to abiotic stress [131].

The phytochromes produced by the plant-beneficial bacteria *Comamonas acidovorans*, *Bacillus* sp., *B. megaterium*, *B. simplex*, *B. subtilis*, and *Paenibacillus polymyxa* induce adventitious root formation in kiwifruit [132]. The formation of new adventitious roots helps plants cope with abiotic stress, including waterlogging, compensating for the root loss caused by hypoxia, especially when the roots are formed closer to the soil surface, where oxygen is more available [112].

Arbuscular mycorrhizal fungi represent a group of microorganisms with strong potential to promote plant growth under waterlogging stress. These microorganisms form symbiosis with the majority of plant groups, including terrestrial plants, plants growing in flooded areas, and water plants [133]. Numerous AMF species are adapted to waterlogging, withstanding low oxygen concentrations, producing plant hormones to stimulate plant adaptations to waterlogging, and storing nutrients in vesicles to help them cope with periodic abiotic stress [133]. Filamentous fungi can transport oxygen from aerated soil to the parts of soil flooded by water, therefore supporting the growth of aerobic bacteria [134]. Moreover, *Funneliformis mosseae* stimulates the expression of aquaporins in trifoliate orange leaves, alleviating waterlogging stress [135].

Moreover, AMF possesses the ability to shape the plant microbiome through the redistribution of plant root exudates [136].

### 3.5. Targeting the Plant Microbiome

Specific plant pathogens vary in their responses to ethylene produced by plants experiencing waterlogging, with ethylene either inhibiting or enhancing pathogen virulence [137]. In terms of the broader microbiome, waterlogging stress is generally known to reduce microbial abundance [138]. The study of the microbial community of rice (*Oryza sativa*) roots has shown that waterlogging reduces microbial richness, decreases microbial diversity, and shifts the rice endophytic bacteria population to more anaerobic species [139]. Plants can, however, adapt to biotic and abiotic stresses through physiological and biochemical adaptations [140]. In addition to the adaptations discussed previously, these responses also include altered root exudation—a process in which plants release a blend of organic compounds, which subsequently influence and are metabolized by the surrounding microorganisms, through their roots. It has been shown that upon infection of durum wheat (*Triticum turgidum* L. var. *durum*) with *Fusarium pseudograminearum*, the plants recruit beneficial microorganisms with antagonistic activity against the tested pathogen [141]. Similarly, plants can adapt their microbiome to abiotic stresses such as drought or waterlogging. This means that a consecutive abiotic stress event will have a less deleterious effect on the plant due to the adapted microbiome [142]. This is especially important since abiotic stresses, such as waterlogging, increase the chance of disease development [143]. This fact can be partially attributed to the decreased plant defenses [5] and increased pathogen virulence in hypoxic conditions [21], as well as to the decreased microbial diversity caused by the environmental change [138,139]. Scheuring and Yu (2012) [144] proposed a mathematical model that suggests that plants recruit their microbiomes to increase microbial competitiveness and diversity. In highly diverse and competitive environments, pathogens are less likely to take over and induce disease [145]. However, in the event of sudden environmental change, such as waterlogging, microbial diversity drastically declines, leaving an open window for pathogen outbreaks [144]. Adaptation to abiotic stress could potentially decrease the negative impact on microbial diversity. However, it would require subjecting plants to stressful conditions, e.g., mild waterlogging, which negatively influences plant growth.

One strategy to protect plants from diseases using microbial communities adapted to changing environmental conditions could be the use of artificial microbial consortia [146]. These multispecies mixtures of microorganisms can provide plant protection in changing environmental conditions better than single microbial strains and, therefore, present a promising approach to enhancing sustainable agricultural production [147]. However, designing effective microbial communities faces several challenges, including inconsistency between in vitro and in vivo effectiveness, compatibility between strains, problems with obtaining stable formulations, and inconsistency in the effectiveness on different soil types and under different weather conditions. The challenges and the solutions proposed to overcome them are summarized by Hossain et al. (2023) [148].

It was reported that certain microbial species help plants recruit other plant-beneficial microorganisms from the environment [149]. In their study, Carlström et al. (2019) [149] showed that certain microbial taxa, such as *Microbacterium*, *Rhodococcus*, *Rhizobium*, and *Sphingomonas*, have superior efficiency in the recruitment of a healthy microbiome. During this study, it was also shown that the early application of these microorganisms increased the chance for the successful shaping of the plant microbiome [149]. This suggests that the early application of beneficial keystone microbial species (species that, disproportionately to their abundance, influence the stability of the whole microbial community [150]) can help develop a beneficial microbiome for increased resistance against biotic and abiotic stresses. More information about the concept of using keystone microbial species in biological plant protection is summarized by Zheng et al. (2021) [151]. Examples of candidate bacterial species that can increase plant fitness through the recruitment of beneficial microorganisms are *Bacillus velezensis* NJAU-Z9 [152] and *Agrobacterium* sp. 10C2 [153]. The inoculation of pepper (*Capsicum annuum* L.) seedlings with *Bacillus velezensis* not only increased the microbial diversity of adult plants but also promoted the growth of plant-beneficial bacteria, including *Bradyrhizobium*, *Chitinophaga*, *Dyadobacter*, *Lysobacter*, *Pseudomonas*, and *Streptomyces*, and it increased plant yield [152]. It has been shown that *Agrobacterium* sp. 10C2—a nodule endophyte—has a positive effect on the growth of common bean, increasing nodulation and bean nutritional quality, partly through the recruitment of plant-beneficial bacteria, e.g., *Bacillus* and *Paenibacillus*, and increased microbial richness [153].

In terms of a more deliberate design of multi-strain consortia tailored for specific applications, significant practical challenges remain. However, the recent acceleration in artificial intelligence development may help overcome existing limitations in analyzing complex natural systems involving numerous factors and interactions [154]. Despite these technological advances, a substantial barrier persists in the form of complicated regulatory processes. Specifically, the registration of microbial preparations containing multiple strains or single strains with biocontrol properties, rather than solely growth-promoting biofertilizers, remains complex and costly [155].

## 4. Conclusions

Plants have developed multiple adaptations to cope with different abiotic stresses, including waterlogging [86]. However, when an environmental change is sudden, plant responses might be rapid and intense, potentially having a severe negative impact on plant growth [62]. On the other hand, a healthy plant microbiome can help plants adapt to these changing environmental conditions by adjusting the proper response and mitigating the negative influence of plant pathogens, which might take advantage of decreased plant defenses [30]. Plant-beneficial microorganisms can help mitigate waterlogging stress through various mechanisms, including the modulation of hormonal responses by reducing ethylene levels or producing plant hormones, the synthesis of metabolites that scavenge reactive oxygen species, and the amendment of the plant microbiome [63]. The microbial taxa that can increase plant tolerance to waterlogging include, inter alia, ACC-deaminase-producing strains of *Pseudomonas*, *Achromobacter*, and *Ochrobactrum* (Table 1). Nevertheless, it is possible that many plant-beneficial microorganisms shown to induce traits associated, inter alia, with increased plant tolerance to waterlogging, such as *Bacillus* and *Trichoderma* species, would also be applicable. Unfortunately, the majority of these strains have not yet been tested in studies aimed specifically at assessing their role in mitigating stress related to waterlogging, indicating an existing knowledge gap. Some of the presented examples include a degree of speculation, as most well-documented cases of the microbial alleviation of waterlogging stress focus on ACC-deaminase-producing strains. Nonetheless, we have chosen to include additional potential mechanisms of action supported by indirect evidence to offer a broader perspective and suggest possible directions for future research.

Although the application of beneficial microorganisms is currently mostly investigated in terms of protection against pathogens or growth stimulation, the mitigation of abiotic stress can be another promising approach to utilizing microorganisms to increase agricultural productivity in changing climate conditions [156]. This is especially important if we consider that plant pathogens require a specific environmental condition for the development of the disease, and properly selected beneficial microorganisms could help to mitigate the stress associated with disease-promoting conditions, in addition to targeting pathogens [23]. Although knowledge about the potential application of microorganisms in agriculture is steadily expanding, more comprehensive, mechanistic, and field-based studies are still required to fully understand and utilize microbial protection against waterlogging in crops [30].

## Figures and Tables

**Figure 1 ijms-26-08034-f001:**
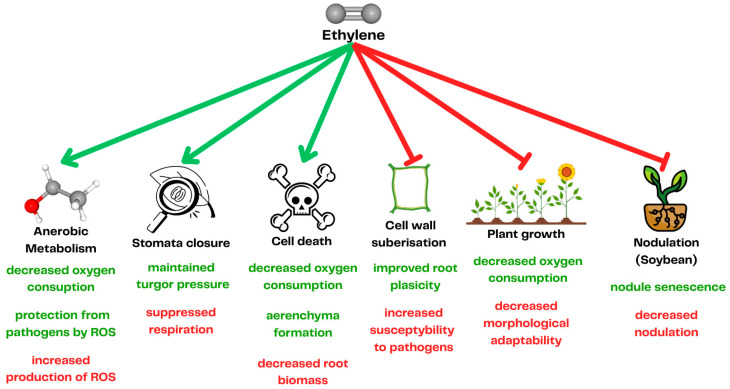
Schematic representation of ethylene’s function during waterlogging. Green arrows represent stimulation, and red arrows represent suppression. The green text highlights positive changes or responses that support plant survival during waterlogging, while the red text indicates the negative impacts of ethylene on plants.

**Figure 2 ijms-26-08034-f002:**
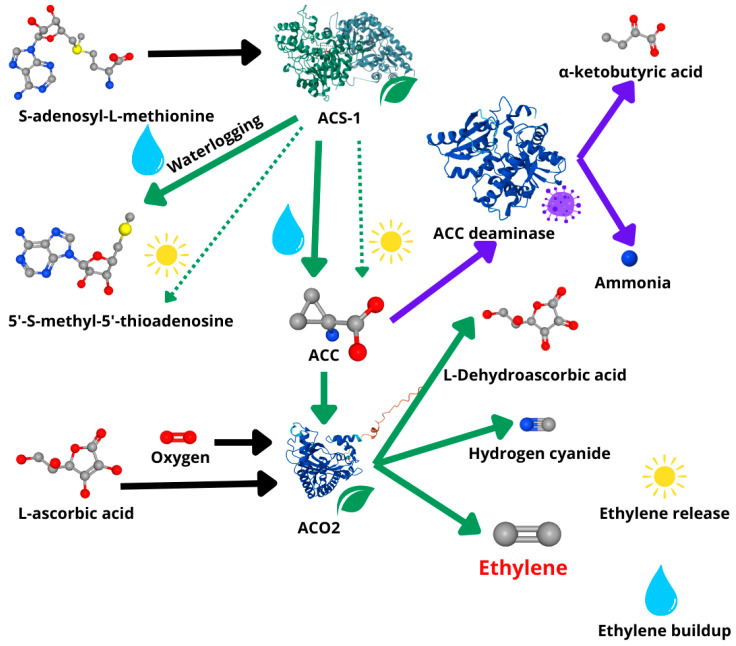
Schematic representation of the ethylene synthesis pathway and how microbial degradation of 1-aminocyclopropane-1-carboxylate (ACC) can lead to a decreased ethylene concentration. Waterlogging (raindrop) stimulates (thick green arrow) the activity of plant 1-aminocyclopropane-1-carboxylate synthase ACS-1, which utilizes S-adenosyl-L-methionine [49], producing the ethylene precursor ACC and 5′-S-methyl-5′-thioadenosine [31]. ACC is later utilized by plant 1-aminocyclopropane-1-carboxylate oxidase 2 ACO2, which transforms ACC, L-ascorbic acid [32], and oxygen [33] into L-Dehydroascorbic acid [36], hydrogen cyanide [50], and ethylene [51]. Bacterial ACC deaminase converts ACC to ammonia [52] and α-ketobutyric acid [53], decreasing the final ethylene concentration and alleviating the negative impact of waterlogging [54]. Waterlogging leads to ethylene buildup due to the reduced escape to the atmosphere. Without waterlogging (sun), ACS-1 is not stimulated and produces ACC at a basal level (dotted green arrow), and ethylene can be released to the atmosphere. The protein structures are downloaded from PDB and cited with appropriate attributions [55]: ACS-1 structure (Apple (*Pyrus malus* L.) ACS-1 P37821 [56]); ACO2 structure (Apple (*Pyrus malus* L.) ACO2 O48882 [28]; bacterial ACC deaminase structure (*Pseudomonas fluorescens* ACC deaminase Q51813 [57]). Black arrows represent the influx of substrates, green arrows represent processed products catalyzed by plant enzymes, and blue arrows represent processes conducted by microbial enzymes.

**Figure 3 ijms-26-08034-f003:**
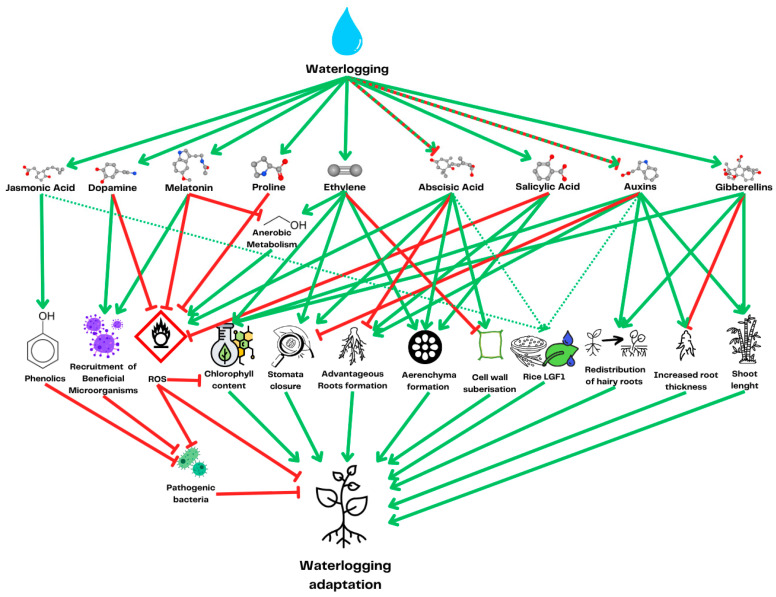
Metabolic and physiological plant adaptations to waterlogging, with the hormones responsible for the induction of changes. Interactions between hormones are omitted for the clarity of the scheme but can be found in Figure 4. The green arrows represent a positive influence on other phytohormone concentrations, while the red arrows represent a negative influence. The dotted green and red lines represent reported cases of variable or context-dependent effects. The dotted narrow green lines leading to Rice LGF1 indicate the prediction of regulation by plant hormones through the sequence analysis of vascular plant one zinc-finger (VOZ) transcription factors, which are responsible for the expression of the rice gene LGF1.

**Figure 4 ijms-26-08034-f004:**
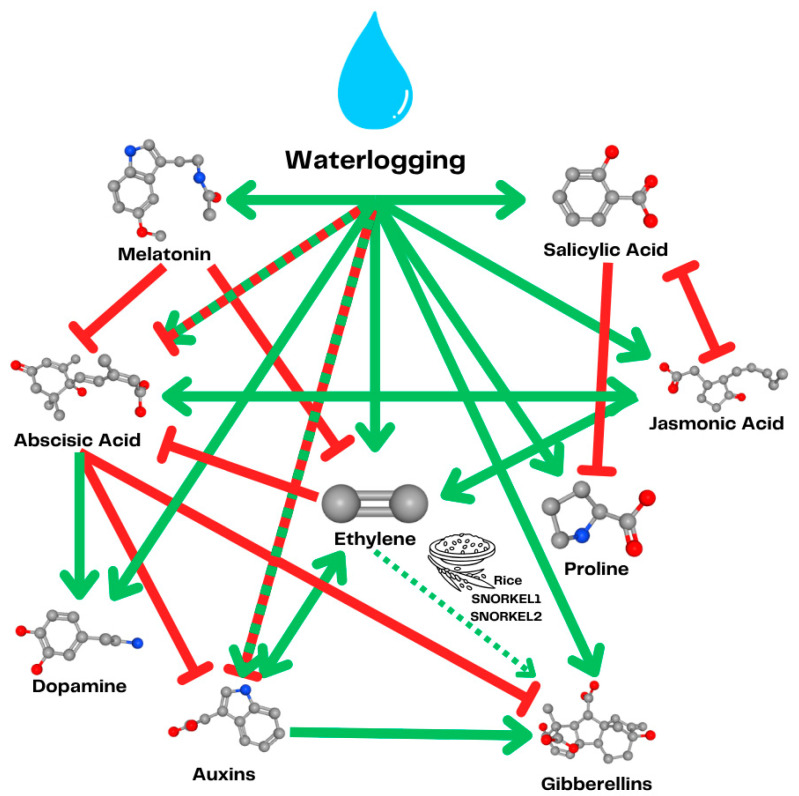
Interplay between plant hormones during waterlogging. Green arrows indicate a positive influence on the concentration of other phytohormones, while red arrows indicate a negative influence. Dotted green and red lines represent reported cases of variable or context-dependent effects, either positive or negative. The dotted green line from ethylene to gibberellins indicates the SNORKL1 and 2 genes from rice, connecting the gibberellin response with ethylene sensing, allowing rice to escape waterlogging and submergence hypoxia caused by internode elongation.

**Figure 5 ijms-26-08034-f005:**
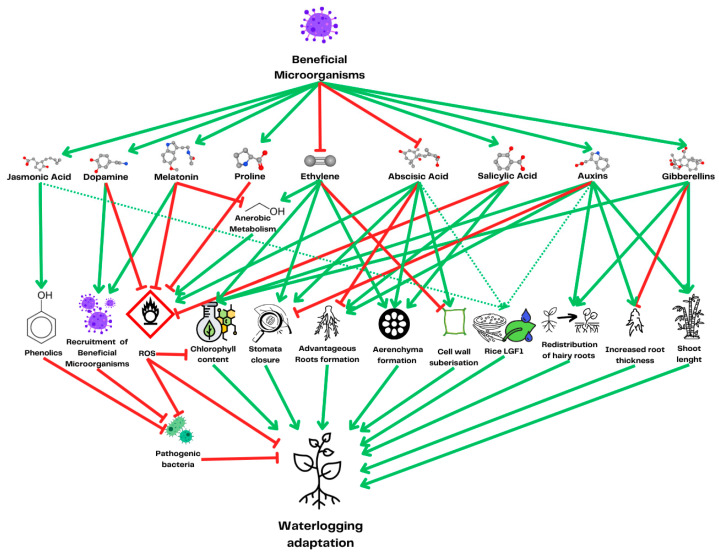
Metabolic and physiological adaptations to waterlogging in plants induced by beneficial microorganisms, along with the hormones responsible for triggering these changes. For clarity, interactions between hormones are not depicted. Green arrows indicate a positive influence on the concentration of other phytohormones, while red arrows indicate a negative influence. The dotted narrow green lines leading to Rice LGF1 indicate the prediction of regulation by plant hormones through the sequence analysis of vascular plant one zinc-finger (VOZ) transcription factors, which are responsible for the expression of the rice gene LGF1.

**Table 1 ijms-26-08034-t001:** List of microbial species experimentally shown to promote plant growth under waterlogging stress through the activity of ACC deaminase.

Species	Model Plant	Observed Response	Additional Properties	Pathogen Antagonism	Reference
*Chromobacter* spp.	wheat (*Triticum aestivum* L.)	SL, SDW, RDW, CH, Apx, N, P, K	Sid, IAA, NH4	not studied	Chandra et al., 2019 [66]
*Pseudomonas* spp.		SL, SDW, RDW, CH, PH, Apx, N, P, K	PS, Sid, IAA, NH4, NF		
*Variovorax paradoxus*		SL, SDW, RDW, CH, PH, N, P, K	Sid, NF		
*Ochrobactrum anthropi*		SL, SDW, RDW, CH, PH, Apx, N, P, K	PS, Sid, IAA, NH4, NF		
*Pseudomonas fluorescens*	rice (*Oryza sativa* L., Cv, Gohar)	RL	Sid, IAA, HCN	not studied	Etasami et al., 2014 [67]
*Achromobacter xylosoxidans*	holy basil (*Ocimum* *tenuiflorum* L.)	SFW, RFW, PL, LN, CH, N, P, Pro, MDA	Sid, IAA, NR, Nod	not studied	Barnawal et al., 2012 [50]
*Serratia ureilytica*		SFW, PL, LN, CH, P	PS, Sid, NR, Nod		
*Herbaspirillum seropedicae*		PL, LN, Pro, N, P,	Sid, IAA, NR		
*Ochrobactrum rhizosphaerae*		SFW, PL, LN, P,	Sid, IAA, NR		
*Pseudomonas putida*	marsh dock (*Rumex palustris* Sm.)	SDW *, SFW, RDW *, RFW *	control of pine wilt disease [51] and drought stress in rapeseed [68]	not studied	Ravanbakhsh et al., 2016 [69]
*Streptomyces* sp.	mung bean (*Vigna radiata* L.)	SL, RL, SFW, RFW, SDW, RDW, CH, LA, SR,	salt stress alleviation in rice [64]	not studied	Jaemsaeng et al., 2018 [65]
*Pseudomonas veronii*	sesame (*Sesamum* *indicum* L.)	SL, RL, FB, DB, CH,	not studied	not studied	Ali et al., 2018 [70]
*Enterobacter cloacae*	tomato (*Lycopersicon esculentum* L.)	SL, SFW, SDW, CH	not studied	Against *Pythium ultimum*; *Rhizoctonia solani*; *Fusarium* *oxysporum* and *Thielaviopsis basicola;* [71]	Grichko and Glick 2001 [72]
*Pseudomonas putida*		SL, SFW, SDW, CH	not studied	not studied	
*Azospirillum brasilense*	radish (*Raphanus sativus* L.)	LN, LA, FB, DB, SC, TRD,	Increased gas exchange in rice roots [73], nitrogen fixation, and denitrification [74]	not studied	Salazar-Garcia et al., 2022 [75]
*Trichoderma asperellum*	wheat (*Triticum aestivum* L.)	CH, SFW, Pro, MDA, SC,	IAA, Pro, Phe, Flav, ROSs	not studied	Rauf et al., 2021 [76]

Observed response: SL—Shoot length, SDW—Shoot dry weight, RDW—Root dry weight, CH—Total chlorophyll, Apx—Ascorbate peroxidase, N—Nitrogen content, P—Phosphate content, K—Potassium content, PH—Total phenol, RL—Root length, SFW—Shoot fresh weight, RFW—Root fresh weight, PL—Plant length, LN—Leaf number, Pro—Proline content, MDA—Malondialdehyde, LA—Leaf area, SR—Survival rate, FB—Fresh biomass, DB—Dry biomass, SC—Stomatal Conductance, TRD—Tuberous root diameter, *—During short 72 h of submergence; Additional mechanisms: Sid—siderophore production, IAA—Indole-3-acetic acid production, NH_4_—Ammonia production, PS—Phosphate solubilization, NF—Nitrogen fixation, HCN-HCN production, NR—Nitrate reduction, Nod—Nodulation, Pro—Proline production, Phe—Polyphenol production, Flav—Flavonoid production, ROSs—Reactive oxygen species scavenging.

## Data Availability

Data sharing is not applicable.

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
