# Peer review of "Microbial Enhancement of Plant Tolerance to Waterlogging: Mechanisms and Interplay with Biological Control of Pathogens"

_ijms, 2025, doi:10.3390/ijms26168034_

Round 1
Reviewer 1 Report
Comments and Suggestions for Authors
General Comments
This review provides a comprehensive synthesis of microbial mechanisms that enhance plant tolerance to waterlogging, with a strong focus on hormonal regulation and reactive oxygen species (ROS) management. The integration of microbial interactions with plant-pathogen dynamics is particularly valuable. However, the manuscript could benefit from clearer structural organization, deeper mechanistic insights, and a more critical analysis of gaps in the field. Below are specific comments to improve the manuscript.
Specific Comments
- Consider specifying key microbial taxa (e.g., Pseudomonas, Bacillus) to highlight actionable insights.
- Briefly mention the practical implications for agriculture (e.g., crop resilience, yield stability).
- The transition between global food demand (lines 1–10) and waterlogging (line 25) could be tightened to avoid redundancy.
- The novelty of this review compared to prior work on plant growth-promoting microorganisms (PGPM) and abiotic stress should be clarified.
- Define "IST" (Induced Systemic Tolerance) at first use (line 50) for broader accessibility.
- Figures 1–3 could be enhanced by including more plant-specific examples (e.g., rice vs. soybean) to illustrate variability in responses.
- The explanation of why anaerobic metabolism increases ROS despite hypoxia (line 130) seems counterintuitive and requires further elaboration.
- The discussion on suberization (line 150) should include trade-offs, such as potential costs to nutrient uptake, for a more balanced critique.
- Table 1 would benefit from an additional column on "Pathogen antagonism" to highlight biocontrol potential.
- In Section 3.2, comparing microbial auxin/IAA production strategies (e.g., tryptophan-dependent pathways) would clarify mechanistic diversity.
- Section 3.5 should expand on microbiome engineering challenges (e.g., strain competition, field efficacy). The discussion on AI (line 700) feels tangential and should be linked to concrete examples.
- Actionable recommendations (e.g., promising microbial combinations for specific crops) should be prioritized.
- Address apparent contradictions, such as the dual role of jasmonates in enhancing stress tolerance while increasing susceptibility to fungi (line 350), and discuss how this might inform strain selection.
- Highlight unresolved questions, such as how microbial communities stabilize under fluctuating waterlogging conditions.
- A table summarizing "Microbial Strains and Their Documented Effects" would provide a quick reference for readers.
Regards
DNA
- The English language requires minor revisions, including
- Avoid overuse of phrases like "what is more" (e.g., line 400); replace with transitions such as "additionally" or "furthermore."
- Defining niche terms (e.g., "keystone strains," line 690) for interdisciplinary readers.
- Ensuring consistency in terminology (e.g., "aerenchyma" is hyphenated inconsistently in line 150 vs. 330).
Author Response
General Comments
This review provides a comprehensive synthesis of microbial mechanisms that enhance plant tolerance to waterlogging, with a strong focus on hormonal regulation and reactive oxygen species (ROS) management. The integration of microbial interactions with plant-pathogen dynamics is particularly valuable. However, the manuscript could benefit from clearer structural organization, deeper mechanistic insights, and a more critical analysis of gaps in the field. Below are specific comments to improve the manuscript.
Thank you for this comment. We have improved the structure, expanded the mechanistic discussion, and strengthened the critical analysis in response to the detailed suggestions from all reviewers.
Specific Comments
Consider specifying key microbial taxa (e.g., Pseudomonas, Bacillus) to highlight actionable insights.
As suggested, we added the information in lines 662-666
Briefly mention the practical implications for agriculture (e.g., crop resilience, yield stability).
We added the information in lines 42-44.
The transition between global food demand (lines 1–10) and waterlogging (line 25) could be tightened to avoid redundancy.
We have removed redundant information from this section, lines 48-53
The novelty of this review compared to prior work on plant growth-promoting microorganisms (PGPM) and abiotic stress should be clarified.
Thank you for this important comment. The topic of plant growth promotion by microorganisms and tolerance to abiotic stress has been heavily investigated, and numerous microbial strains have been proven to induce plant tolerance to various abiotic stress factors. Despite that, not much is known about the induction of plant tolerance against waterlogging, and even less has been achieved in obtaining microorganism-based products for plant growth promotion dedicated intended to enhance plant tolerance to waterlogging.This review aims to summarize current research on microbial alleviation of waterlogging stress and the underlying mechanisms, with the goal of supporting future efforts to identify new microbial strains that can protect plants against waterlogging. This information was, however, missing from the text of the manuscript and therefore it was added in lines 131-137
Define "IST" (Induced Systemic Tolerance) at first use (line 50) for broader accessibility.
We added additional explanation defining IST in lines 69-71.
Figures 1–3 could be enhanced by including more plant-specific examples (e.g., rice vs. soybean) to illustrate variability in responses.
Thank you for this comment. The schemes were intended to present a general idea of the pathways involved in waterlogging response, but this could give a false impression that the response to waterlogging is uniform across different plants. We have added some indication of the differences between different crops' responses to waterlogging using the examples of rice and soybean.
The explanation of why anaerobic metabolism increases ROS despite hypoxia (line 130) seems counterintuitive and requires further elaboration.
Thank you for pointing out the inconsistency in the order of information presented. While the explanation of this phenomenon was originally provided later in the manuscript, we have now moved it earlier, as per your suggestion, to ensure better clarity and coherence.
The discussion on suberization (line 150) should include trade-offs, such as potential costs to nutrient uptake, for a more balanced critique.
Thank you for this important point. We acknowledge the oversight in our reasoning. It stemmed from the fact that, under waterlogging stress, plants typically reduce their metabolic rate, making decreased nutrient uptake seem a relatively minor cost. However, we agree that the previous phrasing could imply that increased suberin content is universally beneficial. We have revised the text accordingly in lines 309–310 to clarify this point.
Table 1 would benefit from an additional column on "Pathogen antagonism" to highlight biocontrol potential.
Thank you for this interesting suggestion. Unfortunately, most of the presented strains have not been tested against plant pathogens, and we could only find that Enterobacter cloacae strains UW4 and CAL2 were tested against Pythium ultimum, Rhizoctonia solani, Fusarium oxysporum, and Thielaviopsis basicola, where the strain CAL2 showed antagonism toward all tested strains and UW4 to none of them. Despite no in vitro antagonisms of UW4 against Pythium and Fusarium, both strains showed statistically significant reduction of disease rate on plant seedlings. We have added the requested column with the information where it was available.
In Section 3.2, comparing microbial auxin/IAA production strategies (e.g., tryptophan-dependent pathways) would clarify mechanistic diversity.
Thank you for this suggestion. We have added more information about the microbial auxin production pathways in lines 426-432.
Section 3.5 should expand on microbiome engineering challenges (e.g., strain competition, field efficacy). The discussion on AI (line 700) feels tangential and should be linked to concrete examples.
Thank you for this suggestion, we have added a discussion on the challenges in microbial engineering in lines 615-620.
Actionable recommendations (e.g., promising microbial combinations for specific crops) should be prioritized.
Thank you for this suggestion. Some general recommendations for designing microbial consortia against waterlogging stress will hugely benefit this manuscript and hopefully help the readers to plan their experiments.
Address apparent contradictions, such as the dual role of jasmonates in enhancing stress tolerance while increasing susceptibility to fungi (line 350), and discuss how this might inform strain selection.
Thank you for this comment. Unfortunately, the current state of knowledge does not allow for a straightforward answer. Although jasmonates are traditionally associated with plant defenses against necrotrophic pathogens, the reality is more complex. Added more discussion in lines 453-458
Highlight unresolved questions, such as how microbial communities stabilize under fluctuating waterlogging conditions.
This is a very important question indeed. Although we know more and more about plant-associated microorganisms, we are still far from fully understanding their ecology. However, the collected experimental data suggest the dynamic fluctuation of stable environments. What it means is that organisms (including bacteria) have to engage in a constant competition where their population dynamically changes during time for the environment to remain stable, like riding on a bike, the faster we go, the easier it is to remain stable. Although this seems to be a promising hypothesis, we still cannot be certain if the stable environment from our perspective can be perceived as stable from the microbial one [1]. Therefore, although this is an excellent topic for scientific discussion and deeper investigation, this review aims to address more technical problems, but according to your recommendation, we have added some background information with suggested literature for readers interested in the topic in lines 628-633.
A table summarizing "Microbial Strains and Their Documented Effects" would provide a quick reference for readers.
Thank you for this excellent suggestion—it was, in fact, our original intention for the summarizing table. However, the vast majority of available data on microbial activity in promoting crop growth under waterlogging conditions relates to strains where the mechanism of action has been identified as ACC deaminase production. Therefore, the most comparable and consistent dataset we could compile is presented in Table 1, which includes ACC deaminase-producing microbial strains with demonstrated plant growth-promoting effects under waterlogging stress. We hope this approach is acceptable and helps clarify the rationale behind this review: although waterlogging is an increasingly relevant stress factor in agriculture, data on effective microbial mitigation strategies remain limited.
Comments on the Quality of English Language
We apologize for the mistakes in the English language. We have revised the manuscript to correct spelling mistakes and reduce unnecessary repetitions.
The English language requires minor revisions, including
Avoid overuse of phrases like "what is more" (e.g., line 400); replace with transitions such as "additionally" or "furthermore."
Corrected accordingly.
Defining niche terms (e.g., "keystone strains," line 690) for interdisciplinary readers.
Thank you for spotting this issue. This term is not commonly used in microbiology, as it was only recently adopted from the ecology of multicellular organisms and may therefore require clearer explanation. The meaning of the term was explained.
Ensuring consistency in terminology (e.g., "aerenchyma" is hyphenated inconsistently in line 150 vs. 330)
Corrected.
Reviewer 2 Report
Comments and Suggestions for Authors
The review synthesizes how plant-beneficial microorganisms mitigate waterlogging stress through mechanisms such as ACC deaminase-mediated ethylene reduction, reactive oxygen species (ROS) scavenging, hormonal modulation (e.g., auxin, melatonin), and root architectural adaptations (e.g., aerenchyma formation). It further links these abiotic stress-alleviating traits to enhanced pathogen resistance, proposing a dual role for microbial interventions in climate-resilient agriculture. Key examples include Pseudomonas and Bacillus, which improve plant fitness under hypoxia while suppressing pathogens like Pectobacteriaceae. While the review advances a compelling integrative framework for microbial-mediated stress resilience, its limitations-particularly mechanistic gaps and field validation-constrain its utility for both basic research and agricultural innovation. For example, ACC deaminase activity and hormonal crosstalk are emphasized, molecular mechanisms-such as ethylene signaling networks interfacing with ROS homeostasis or auxin transport dynamics-remain underexplored. Most cited evidence derives from lab/greenhouse experiments, with minimal field validation. Others, there is a problem of inconsistent formatting of references (such as missing page numbers or volume numbers in some citations).
Author Response
The review synthesizes how plant-beneficial microorganisms mitigate waterlogging stress through mechanisms such as ACC deaminase-mediated ethylene reduction, reactive oxygen species (ROS) scavenging, hormonal modulation (e.g., auxin, melatonin), and root architectural adaptations (e.g., aerenchyma formation). It further links these abiotic stress-alleviating traits to enhanced pathogen resistance, proposing a dual role for microbial interventions in climate-resilient agriculture. Key examples include Pseudomonas and Bacillus, which improve plant fitness under hypoxia while suppressing pathogens like Pectobacteriaceae. While the review advances a compelling integrative framework for microbial-mediated stress resilience, its limitations-particularly mechanistic gaps and field validation-constrain its utility for both basic research and agricultural innovation. For example, ACC deaminase activity and hormonal crosstalk are emphasized, molecular mechanisms-such as ethylene signaling networks interfacing with ROS homeostasis or auxin transport dynamics-remain underexplored. Most cited evidence derives from lab/greenhouse experiments, with minimal field validation. Others, there is a problem of inconsistent formatting of references (such as missing page numbers or volume numbers in some citations).
We would like to thank the reviewer for the insightful and constructive comments. Your perspective helped us identify key gaps in our literature review, particularly regarding auxins (lines 406-412) and reactive oxygen species homeostasis (lines 165-179 ), which we have now expanded based on your and the other reviewers’ suggestions. As you rightly noted, some information is missing—not due to oversight, but because the area of microbial induction of plant tolerance to waterlogging remains largely underexplored. In particular, there is a clear lack of field-based studies addressing this issue. We hope that our review draws attention to these gaps and supports the formulation of new research hypotheses in this emerging area. Given the increasing threat of waterlogging under climate change scenarios, we believe that advancing microbial solutions may play a crucial role in mitigating its negative impact on agriculture.
Reviewer 3 Report
Comments and Suggestions for Authors
This review manuscript ‘by Maciag and Krzyźanowska’ tries to describe the potential ‘effect and mode’ of biostimulants (PGPR) toward water stresses. However, a number of areas in this manuscript are still speculative instead of collecting/reviewing fast and current research results.
- A manuscript should define ‘PGPM’ (vs. a widely used term, PGPR)? Provide the exemplary strain of PGPM that doesn’t belong to PGPR?
- Define ‘waterlogging’. Does it describe if or not plant aerial parts are submerged? If so, how much? Do plants genetically/metabolically respond similarly/differently when only roots are water-saturated vs. roots and aerial tissues are water-submerged? If same or different, what and how? In addition (L126), what about the hydroponic system? – do plants under the hydroponic system activate water-stress-like mechanisms?
- L66, “some mechanisms underlying the enhancement of plant tolerance to abiotic stress closely resemble those involved in inducing plant resistance to biotic stress, there are, however, key differences worth emphasizing”. Explain it (examples?).
- L68-82, it seems that authors referred to “immunity” rather than “resistance”. For instance (in Arabidopsis and syringae system), 1 log reduction of bacteria growth (and/or HR activation etc.) is considered as “resistance” – rather than complete ‘immunity’ of plants against pathogen infections.
- How do Pectobacteriaceae sense the decrease plant defenses and favorable environmental conditions? Genetically? Chemotactically?
- What are conserved defense mechanisms between abiotic and biotic responses?
- “On the other hand, the resulting thinner plant cell walls can make it easier for pathogens to invade [17]”. This is an example that authors (throughout a manuscript) mixed using the examples of biotrophic and necrotrophic pathogens, and soilborne and airborne pathogens. Their infection (invade) pathways (and associated hormone signaling) are largely different from each other.
- “Therefore, although ethylene plays a key role in plant adaptation to waterlogging, it requires regulation by other plant hormones to mitigate the negative impacts of ethylene accumulation”. Provide/explain major studies to directly concur the statement. There are a large number of ethylene-associated mutant and transgenic plants as well as other hormone-associated mutant and transgenic plants.
- Chapter 2 is largely speculative.
- L152-193, it supports little of waterlogging stress.
- It .is unclear how the cited paper explains “To minimize ROS toxicity, plants have to reduce their metabolic rate during the flooding period [58]”. Wonder a major mechanism of ROS detoxification employ enzymatic processes?
- Lysogenous to Lysigenous, Are we talking about Lysienous aerenchyma formation for wetland and upland plants? ET could work differently in these pathways, Is there any experiment showing of PGPR induce lysigenous aerenchyma?
- L284- 305. A major point is confused?
- “IAA-producing Klebsiella variicola AY13 increases soybean tolerance to waterlogging stress and increases shoot length, root length, and chlorophyll content of the submerged plants [96].” Can we directly correlate the Klebsiella variicola-induced waterlogging-tolerance with IAA”? Tested with isogenic line without IAA production?
- Targeting reactive oxygen species. Again, is this for upland plants?
A manuscript is not conclusive 1) how effective PGPR (or PGPM) is against the water stresses of which kinds, and 2) what is mechanism of PGPR (PGPR)-mediated water tolerance (esp., the role of ethylene, ROS and their crosstalk are still elusive).
Author Response
This review manuscript ‘by Maciag and Krzyźanowska’ tries to describe the potential ‘effect and mode’ of biostimulants (PGPR) toward water stresses. However, a number of areas in this manuscript are still speculative instead of collecting/reviewing fast and current research results.
We would like to thank the reviewer for all the helpful comments, which helped us improve our manuscript. This review addresses the utilization of microorganisms to alleviate waterlogging stress; however, the examples of microorganisms with experimentally proven ability to increase plant tolerance to waterlogging are scarce and almost exclusively describe one mechanism, namely the production of ACC deaminase. Therefore, in this review, we have also included examples of microbial induculants showing traits associated with increased tolerance to waterlogging, even if originally studied with a different research question. We agree that it is partially speculative and needs to be experimentally confirmed; however, it provides some directions in the context of the current lack of data. This review aims, however, to summarize what has been achieved so far and indicate possible future directions for studies dedicated to finding microorganisms that alleviate waterlogging stress, which is predicted to play a more and more significant role in agriculture due to climate change.
- A manuscript should define ‘PGPM’ (vs. a widely used term, PGPR)? Provide the exemplary strain of PGPM that doesn’t belong to PGPR?
PGPR is an older and more widely used term that. However, it describes plant growth-promoting rhizobacteria, not including fungi nor phyllosphere bacteria. Therefore, the term PGPM, which is broader and also well-established, was used in this work. We added the explanation in line 59.
- Define ‘waterlogging’. Does it describe if or not plant aerial parts are submerged? If so, how much? Do plants genetically/metabolically respond similarly/differently when only roots are water-saturated vs. roots and aerial tissues are water-submerged? If same or different, what and how? In addition (L126), what about the hydroponic system? – do plants under the hydroponic system activate water-stress-like mechanisms?
We have added an explanation of waterlogging in line 117. Waterlogging describes the process of water saturation of soil. When the water reaches above the soil level, the process is called submergence. Plants respond similarly to waterlogging and submergence since it relies on ethylene level sensing; however, there are some differences in the response of waterlogging/submergence adapted species like rice, which is described in more detail in sections 2.1 and 2.2. Plants in hydroponic systems can also suffer from hypoxia; however, due to the stable growth conditions, the change in water level does not cause the oxidative burst, and hydroponically grown plants develop thicker roots, which help plants to withstand the conditions of lower oxygen availability. Despite that, many plants do not require additional aeration.
- L66, “some mechanisms underlying the enhancement of plant tolerance to abiotic stress closely resemble those involved in inducing plant resistance to biotic stress, there are, however, key differences worth emphasizing”. Explain it (examples?).
Thank you for this comment. We are sorry to have oversimplified this issue. We have added the examples supporting this statement in lines 72-79
- L68-82, it seems that authors referred to “immunity” rather than “resistance”. For instance (in Arabidopsis and syringae system), 1 log reduction of bacteria growth (and/or HR activation etc.) is considered as “resistance” – rather than complete ‘immunity’ of plants against pathogen infections.
We are sorry for this oversimplification. We have revised this fragment to better distinguish between resistance and tolerance, underlining that resistance is a more qualitative feature while tolerance is a more quantitative.
- How do Pectobacteriaceae sense the decrease plant defenses and favorable environmental conditions? Genetically? Chemotactically?
Thank you for this comment, we have added the missing information in lines 105-107.
- What are conserved defense mechanisms between abiotic and biotic responses?
The key element of similarity is the crosstalk between the hormonal signaling of those two types of plants. Abscisic acid, jasmonic acid, salicylic acid, and ethylene are major plant hormones playing a pivotal role in the development of stress response to both abiotic and biotic stress. While the abscisic acid is most traditionally connected to drought and salinity stress response, jasmonic acid is associated with the response to necrotrophic pathogens, and salicylic acid to biotrophic pathogens. However, abscisic acid can also increase plant resistance to some pathogens, and jasmonic acid and salicylic acid are important for the development of plant defenses against abiotic stress. This can be partially explained by the fact that, for example, increased cell wall suberisation can increase plant defenses against pathogen invasion, but also reduce radial oxygen loss during waterlogging.
- “On the other hand, the resulting thinner plant cell walls can make it easier for pathogens to invade [17]”. This is an example that authors (throughout a manuscript) mixed using the examples of biotrophic and necrotrophic pathogens, and soilborne and airborne pathogens. Their infection (invade) pathways (and associated hormone signaling) are largely different from each other.
Thank you for this comment. We might tend to mix the necrotrophic and biotrophic invasion since probably the best example of pathogens that are closely related to root hypoxia caused by waterlogging are soft rot Pectobacteriaceae, which have a dual strategy, depending on the infection stage. However, waterlogging is indeed most beneficial for the necrotrophic invasion of the host roots. The examples from airborne pathogens are included to demonstrate the volatile communication between the host and pathogens, which is better understood for the above-ground parts of the plant due to technical limitations.
- “Therefore, although ethylene plays a key role in plant adaptation to waterlogging, it requires regulation by other plant hormones to mitigate the negative impacts of ethylene accumulation”. Provide/explain major studies to directly concur the statement. There are a large number of ethylene-associated mutant and transgenic plants as well as other hormone-associated mutant and transgenic plants.
Thank you for this comment. We added the example for the studies indicating that the good adaptation to waterlogging requires additional hormonal regulation (lines 193-195).
- Chapter 2 is largely speculative.
We agree that Chapter 2 includes a degree of speculation. This results from the current limitations in available experimental data, as most well-documented cases of microbial alleviation of waterlogging stress focus on ACC deaminase-producing strains. Nonetheless, we chose to include additional potential mechanisms of action, supported by indirect evidence, to offer a broader perspective and suggest possible directions for future research. We acknowledge that these mechanisms require further experimental validation.
- L152-193, it supports little of waterlogging stress.
This paragraph refers to the regulation of ethylene signaling, a key waterlogging sensing and signaling molecule, and its connection to other plant hormones responsible for the induction of metabolic and physiological adaptations. Line 212-230 - It .is unclear how the cited paper explains “To minimize ROS toxicity, plants have to reduce their metabolic rate during the flooding period [58]”. Wonder a major mechanism of ROS detoxification employ enzymatic processes?
The presented papers show the differences in metabolism rate of 3 different plant species differing in adaptability to waterlogging, showing that the major difference in the most tolerant rice is conditioned by its ability to enter metabolic dormancy and maintain efficient photosynthesis even in anoxic conditions. We have added information about enzymatic ROS scavenging, which plays an important role in the scavenging of ROS at the initial stages of hypoxia. - Lysogenous to Lysigenous, Are we talking about Lysienous aerenchyma formation for wetland and upland plants? ET could work differently in these pathways, Is there any experiment showing of PGPR induce lysigenous aerenchyma?
The lysogenous aerenchyma formation under waterlogging conditions is similarly regulated by both wetland and dryland species. However, waterlogging-adapted species can also produce lisogenous aerenchyma without waterlogging-induced accumulation of ethylene. We have added the requested information to the manuscript's main text. Yes, aerenchyma formation can be induced by PGPR e.g., in maize by Paenibacillus lentimorbus, inducing its tolerance to nutrient deficiency stress information (in lines 520-521). We could not find any direct information on the PGPR induction of aerenchyma linked to increased tolerance to waterlogging. Despite that, there is multiple experimental evidence on the positive impact of aerenchyma formation on waterlogging tolerance. - L284- 305. A major point is confused?
Rephrased to increase clarity. - “IAA-producing Klebsiella variicola AY13 increases soybean tolerance to waterlogging stress and increases shoot length, root length, and chlorophyll content of the submerged plants [96].” Can we directly correlate the Klebsiella variicola-induced waterlogging-tolerance with IAA”? Tested with isogenic line without IAA production?
In these experiments, the authors have not confirmed that it is only the production of IAA that is responsible for increasing tolerance to waterlogging. Multiple experimental data indicate that IAA production by PGPR is a key mechanism of plant growth promotion and induction of tolerance against different abiotic stresses, but not against waterlogging. Here we present an example of a microorganism increasing plant tolerance to waterlogging, indicating the possible mode of action suggested by the authors. - Targeting reactive oxygen species. Again, is this for upland plants?
This review concerns mostly upland crop species; however, while writing about adaptations to waterlogging conditions the rice is a perfect example of a model organism of crop species that possesses multiple adaptations against waterlogging.
A manuscript is not conclusive 1) how effective PGPR (or PGPM) is against the water stresses of which kinds, and 2) what is mechanism of PGPR (PGPR)-mediated water tolerance (esp., the role of ethylene, ROS and their crosstalk are still elusive
We would like to thank you for this comment, and would be happy to provide conclusive information about the effectiveness of PGPM on increasing waterlogging tolerance of crops. Unfortunately, the presented data are too scarce to provide such conclusions. With this review, we wanted to summarize what has been achieved so far in this topic to help scientists direct their future research. We believe that in the future, climate change will cause the losses in agriculture caused by waterlogging to increase, and therefore, we need to be prepared to protect crops against this devastating abiotic stress. We acknowledge that some parts of the manuscript may be considered speculative; however, such content is included intentionally, based on current predictions and the lack of comprehensive data. The process of analyzing the literature for this review has helped us identify key gaps in the existing research, and we hope it will also serve as a useful resource for future readers and researchers in this field.
We hope that, based on your comments and those of the other reviewers, we have significantly improved the clarity of this manuscript.
Round 2
Reviewer 3 Report
Comments and Suggestions for Authors
The responses to the earlier revision is not satisfactory; as those are not clear how/what were changed, and/or where (incorrect line #) were revised?
Based on authors’ responses and the revised manuscript, this review manuscript should be considered to reformat for a shorter review (2 to 3 pages), especially as a reviewer and authors both agree that there is little study done for the effect of PGPM to waterlogging stress.
Most review articles (covering PGPR/PGPM activities toward environmental stresses) similarly discuss how those primary and secondary metabolites, derived from PGPR/PGPM, works to (i) assist plant hormone/metabolite defense signaling, and (ii) protect plants from a variety of stresses – with more or less the same mechanism.
Although overall manuscript structure could be reformatted, authors should clarify (e.g., from Table 1) how many PGPM demonstrate waterlogging tolerance (toward which plants)? And discuss their potential roles and activities.
Author Response
The responses to the earlier revision is not satisfactory; as those are not clear how/what were changed, and/or where (incorrect line #) were revised?
We have included responses to all your comments. Could you let us know which of our responses are still unsatisfactory? The responses have information about the localization of changes in the manuscript text using new vers lines. We have uploaded the version of the manuscript with tracked changes. Additionally, we have also uploaded the version without tracked changes as the text is clearer to read for some reviewers, but the line numbers correspond to the version with tracked changes (which is required by IJMS). Due to the extensive changes during the first round of revisions, the text formatting has been largely disrupted, which was corrected by MDPI, but has led to some shift in lines of the incorporated changes. Therefore, we found the changes in the properly formatted version of the manuscript and corrected the line numbers in our response.
Based on authors’ responses and the revised manuscript, this review manuscript should be considered to reformat for a shorter review (2 to 3 pages), especially as a reviewer and authors both agree that there is little study done for the effect of PGPM to waterlogging stress.
We agree that the presented area is largely understudied; however, this does not mean that there are no data relevant to the subject. Since reviews are mostly read by scientists, we believe that highlighting the research gaps in the field is as important as summarizing the findings of the field. We could rearrange the whole manuscript to concentrate on the summary of examples of microorganisms currently used to alleviate waterlogging stress in crops, but it would be largely against the other reviewers' suggestions, who have already been satisfied by our implemented changes that further expanded the manuscript.
Most review articles (covering PGPR/PGPM activities toward environmental stresses) similarly discuss how those primary and secondary metabolites, derived from PGPR/PGPM, works to (i) assist plant hormone/metabolite defense signaling, and (ii) protect plants from a variety of stresses – with more or less the same mechanism.
We fully agree with the statement, most of the presented data on the mechanism of microbial alleviation of waterlogging stress concentrate on the reduction of ethylene levels by ACC deaminase. Therefore, we have presented such strains with known mechanisms of waterlogging stress alleviation in Table 1. However, we hope that new studies will help to identify new mechanisms of waterlogging stress alleviation. Similarly, in biological plant protection against plant diseases, most of the research is dedicated to direct antibiosis; however, new studies find new promising mechanisms that can be used to more efficiently protect plants against diseases. Therefore, we hope that the presented literature analysis will help other scientists to look for such new mechanisms.
Although overall manuscript structure could be reformatted, authors should clarify (e.g., from Table 1) how many PGPM demonstrate waterlogging tolerance (toward which plants)? And discuss their potential roles and activities.
Table 1 presents only the microorganisms inducing plant tolerance to waterlogging by known mechanisms, the used model plant, observed plant response, and additional properties of the strain found in the original manuscript or other research using the same strains. We could also prepare a table containing the list of microorganisms alleviating waterlogging stress without an identified mode of action, but it would further lengthen the manuscript.